# On Inductive Biases That Enable Generalization of Diffusion Transformers

Jie An[1,2][*], De Wang[1], Pengsheng Guo[1], Jiebo Luo[2], Alexander G. Schwing[1]
[1]Apple,   [2]University of Rochester
{jan6,jluo}@cs.rochester.edu
{de_wang,pengsheng_guo,ag_schwing}@apple.com

## Abstract

Recent work studying the generalization of diffusion models with locally linear UNet-based denoisers reveals inductive biases that can be expressed via geometry-adaptive harmonic bases. For such locally linear UNets, these geometry-adaptive harmonic bases can be conveniently visualized through the eigen-decomposition of a UNet's Jacobian matrix. In practice, however, more recent denoising networks are often transformer-based, *e.g.*, the diffusion transformer (DiT). Due to the presence of nonlinear operations, similar eigen-decomposition analyses cannot be used to reveal the inductive biases of transformer-based denoisers. This motivates our search for alternative ways to explain the strong generalization ability observed in DiT models. Investigating a DiT's pivotal attention modules, we find that locality of attention maps in a DiT's early layers are closely associated with generalization. To verify this finding, we modify the generalization of a DiT by restricting its attention windows and observe an improvement in generalization. Furthermore, we empirically find that both the placement and the effective attention size of these local attention windows are crucial factors. Experimental results on the CelebA, ImageNet, MSCOCO, and LSUN data show that strengthening the inductive bias of a DiT can improve both generalization and generation quality when less training data is available. Source code is available at `https://github.com/DiT-Generalization/DiT-Generalization`.

## 1   Introduction

Diffusion models have achieved remarkable success in visual content generation. Their training involves approximating a distribution in a high-dimensional space from a limited number of training samples–a task that is demanding due to the curse of dimensionality. Nonetheless, recent diffusion models (Song et al., 2020; Ho et al., 2020) learn to generate high-quality images (Nichol et al., 2021; Dhariwal & Nichol, 2021; Saharia et al., 2022; Rombach et al., 2022; Chen et al., 2023, 2024a) and even videos/audio (Singer et al., 2022; Ho et al., 2022; Girdhar et al., 2023; Blattmann et al., 2023; OpenAI, 2024; Cheng et al., 2025) using relatively few samples when compared to the dimensionality of the underlying space. This indicates that diffusion models exhibit powerful inductive biases (Wilson & Izmailov, 2020; Goyal & Bengio, 2022; Griffiths et al., 2024) that promote effective generalization. What exactly are these inductive biases? Answering this question is crucial for understanding the behavior of diffusion models and their generalization.

Recent work by Kadkhodaie et al. (2024) on locally linear single-channel UNet-based diffusion models reveals that the strong generalization of UNet-based denoisers is driven by inductive biases that can be expressed via a set of geometry-adaptive harmonic bases (Mallat et al., 2020). For a UNet that has been modified to be locally linear, such harmonic bases can be extracted via the eigenvectors

---

[*]Work done during internship at Apple.

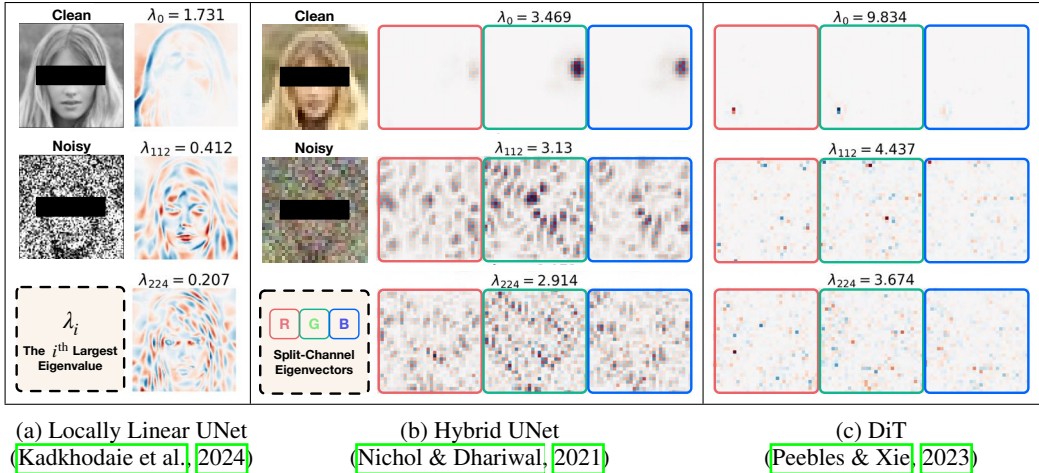

(a) Locally Linear UNet
(Kadkhodaie et al., 2024)

(b) Hybrid UNet
(Nichol & Dhariwal, 2021)

(c) DiT
(Peebles & Xie, 2023)

Figure 1: Jacobian eigenvectors of (a) a locally linear one-channel UNet, (b) the hybrid UNet introduced in improved diffusion (Nichol & Dhariwal, 2021), and (c) a DiT (Peebles & Xie, 2023). Kadkhodaie et al. (2024) find that the generalization of UNet-based diffusion models is driven by geometry-adaptive harmonic bases (a), which exhibit oscillatory patterns whose frequency increases as the eigenvalue $\lambda_k$ decreases. For hybrid UNets (Nichol & Dhariwal, 2021), due to the inclusion of non-linear operations such as softmax in transformer blocks and normalization layers in both transformer and convolutional layers, the harmonic bases extracted from their split-channel eigenvectors (b) do not adapt well to the input geometry, though oscillatory patterns still persist. In contrast, the harmonic bases completely disappear in a DiT (Peebles & Xie, 2023) as shown in (c), indicating that the eigen-decomposition analysis is no longer valid for transformer-based DiTs. This observation motivates us to investigate the inductive biases of a DiT that enable its generalization. The RGB channels of the split-channel eigenvectors are outlined with red, green, and blue boxes, respectively. All models operate directly in the pixel space without the patchify operation.

of the denoiser's Jacobian matrix, as shown in Fig. 1(a). Extending the eigen-decomposition analysis of Kadkhodaie et al. (2024) to more complex, classic multi-channel UNets shows that geometry-adaptive harmonic bases become harder to observe. As illustrated in Fig. 1(b), these eigenvectors do not adapt well to the input geometry, although oscillatory patterns whose frequencies increase as the eigenvalues $\lambda_k$ decrease still exist. This is because modern UNets (Nichol & Dhariwal, 2021) adopt hybrid architectures that incorporate several transformer layers, where the softmax in attention and normalization layers, present in both convolutional and transformer blocks, degrade the network's local linearity. For DiT, the eigenvectors of its Jacobian matrix show no geometry-adaptive harmonic bases, as shown in Fig. 1(c). This does not necessarily imply that a DiT fails to capture geometry-adaptive harmonic structures, but rather suggests that the eigen-decomposition analysis of Kadkhodaie et al. (2024) isn't applicable for probing the inductive bias of DiT models. Motivated by this issue, we seek alternative approaches to address the question: what inductive biases enable the strong generalization ability of DiTs?

Answering this question is particularly important because of the recent growing adoption of DiTs (Chen et al., 2024b; Esser et al., 2024; Cheng et al., 2025; Chen et al., 2025), partly for its observed performance at scale (Peebles & Xie, 2023). In a new study in this paper, using the PSNR gap (Kadkhodaie et al., 2024) as a metric to evaluate the generalization of diffusion models, we confirm that a DiT indeed exhibits better generalization than a UNet with the same FLOPs. Yet, this observation alone doesn't reveal the inductive biases which enable generalization.

The generalization mechanism of a DiT can be determined by inductive biases introduced by the diffusion model theory, training objectives, target optimal score functions, and network architectures. Prior works (Zhang et al., 2024; Niedoba et al., 2024; Li et al., 2024; Wang & Vastola, 2024) reveal that the inductive biases enabled by diffusion model theory, training objectives, and target optimal score functions can be similar between a UNet and a DiT. However, the inductive bias driven by the model architecture, differs between a UNet and a DiT, potentially due to the self-attention (Vaswani, 2017) dynamics which are pivotal in DiT models but not in UNets. In a self-attention layer, the

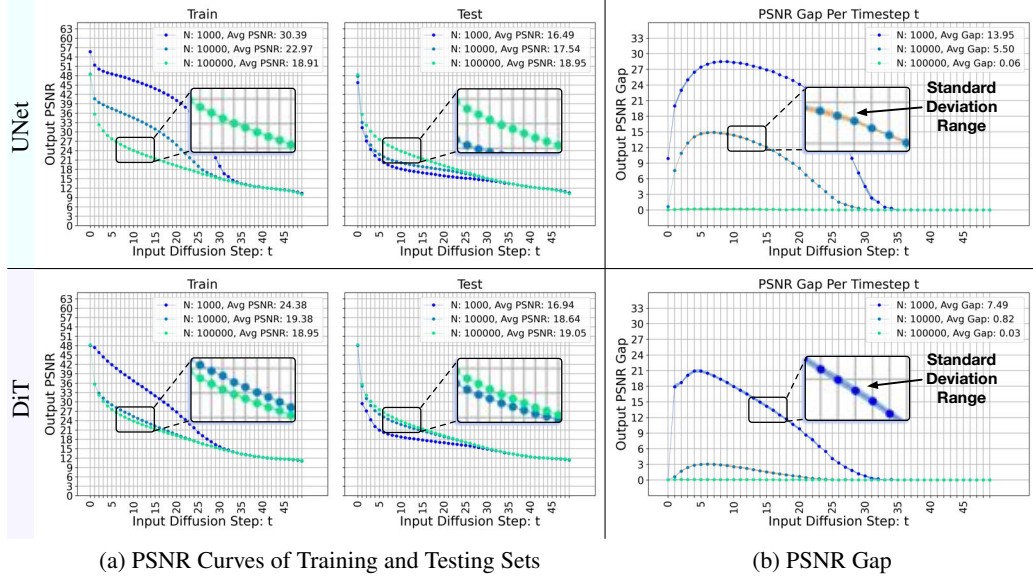

(a) PSNR Curves of Training and Testing Sets          (b) PSNR Gap

Figure 2: PSNR (a) and PSNR gap (b) comparison between a UNet and a DiT with the same FLOPs for different training image quantities ($N$). All curves are averaged over three training runs using different dataset shuffles. The standard deviations, illustrated by the curve shadows in the zoomed-in windows, are negligible, indicating minimal variation.

attention map, derived from the multiplication of query and key matrices, determines how the value matrix obtained from input tensors influences output tensors. To shed some light, we analyze the attention maps of a DiT and find that locality of the attention maps is closely tied to its generalization ability. Specifically, the attention maps of a DiT trained with insufficient images, *i.e.*, a DiT with weak generalization, exhibit a more position-invariant pattern, especially in early layers: the output tokens of a self-attention layer are largely influenced by a certain combination of input tensors, irrespective of their positions. In contrast, the attention maps of a DiT trained with sufficient images, which demonstrates strong generalization, exhibit a sparse diagonal pattern. This indicates that each output token is primarily influenced by its neighboring input tokens. This analysis provides insight into how the generalization ability of DiTs can be modified, if necessary, such as when only a small number of training images are available.

If the above finding is true, restricting the attention window in self-attention layers should permit us to modify a DiT's generalization. Indeed, we find that employing local attention windows (Beltagy et al., 2020; Hassani et al., 2023) is effective. A local attention window restricts the dependence of an output token on its nearby input tokens, thereby promoting the locality of attention maps. In addition, the placement of attention window restrictions within the DiT architecture and the effective size of attention windows are critical factors to steer a DiT's generalization. Our experiments show that placing attention window restrictions in the early attention layers of the DiT architecture has most impact. Results on CelebA (Liu et al., 2015), ImageNet (Deng et al., 2009), MSCOCO (Lin et al., 2014), and LSUN (Yu et al., 2015) (bedroom, church, tower, bridge) data reveal that applying attention window restrictions modifies generalization, as reflected by a reduced PSNR gap. We also observe improved FID (Heusel et al., 2017), Inception Score (IS) (Barratt & Sharma, 2018), and FD-DINOv2 (Oquab et al., 2023) when training with insufficient data, confirming that a DiT's generalization can be successfully modified through attention window restrictions.

In summary, the contributions of this paper include the following: 1) We identify the locality of attention maps as a key inductive bias contributing to the generalization of a DiT, and 2) we demonstrate how to control this inductive bias by incorporating local attention windows into early layers of a DiT. Enhancing the locality in attention computations effectively modifies a DiT's generalization, resulting in a lower PSNR gap and improved FID, IS, and FD-DINOv2 scores when insufficient training images are available for training.

## 2 Inductive Bias Analysis of Diffusion Models

Diffusion models are designed to map a Gaussian noise distribution to a dataset distribution. To achieve this, diffusion models can be formulated to estimate the noise $\epsilon$ that was used to compute the corrupted image $x_t$ by perturbing the training sample $x_0$ following a noise schedule depending on step $t$. The loss function of diffusion model training hence reads as follows:

$$\mathcal{L} = \mathbb{E}_{x_0, \epsilon, t} \left[ \| \epsilon - \epsilon_\theta (x_t, t) \|_2^2 \right]. \tag{1}$$

Here, $\epsilon_\theta(\cdot)$ represents the backbone network with trainable parameters $\theta$, which plays a crucial role in diffusion model generalization. In this section, we first compare the generalization ability of a DiT (Peebles & Xie, 2023) and a UNet (Nichol & Dhariwal, 2021), two of the most popular diffusion model backbones. Subsequently, we investigate the inductive biases that drive their generalization.

### 2.1 Comparing DiT and UNet Generalization

We compare the generalization of pixel-space DiT and UNet[2] using as a metric the PSNR gap proposed by Kadkhodaie et al. (2024). The PSNR gap at a diffusion step $t$, denoted as $\mathrm{Gap}\,(t)$, is the zero-truncated difference between the training set PSNR and the testing set PSNR at step $t$:

$$\mathrm{Gap}\,(t) = \max \left( \mathrm{PSNR}_{\text{train}}\,(t) - \mathrm{PSNR}_{\text{test}}\,(t), 0 \right), \tag{2}$$

where $\mathrm{PSNR}_{\text{train}}\,(t)$ and $\mathrm{PSNR}_{\text{test}}\,(t)$ are obtained following Kadkhodaie et al. (2024). To elaborate, given $K$ images from either training or testing set, we first feed noisy images at step $t$ to diffusion models and obtain the estimated noise $\hat{\epsilon}$. Next, we compute the one-step denoising result $\hat{x}_0$ via

$$\hat{x}_0 = x_t - \sigma_t \hat{\epsilon}, \tag{3}$$

where $\sigma_t$ is defined by the diffusion model noise schedule. Finally, we derive the training and testing PSNRs at diffusion step $t$ as follows:

$$\mathrm{PSNR}\,(t) = \frac{1}{K} \sum_{k=1}^{K} \left( 10 \cdot \log \left( \frac{M^2}{\mathrm{MSE}\left( \hat{x}_0^k, x_0^k \right)} \right) \right). \tag{4}$$

Here, $\hat{x}_0^k$ denotes the estimate of image $k$, obtained by using Eq. (3), $M$ denotes the intensity range of $x_0$, which is set to 2 since images are normalized to $[-1, 1]$. $K$ is set to 300 following the PSNR gap computation of Kadkhodaie et al. (2024).

Turning to diffusion model backbones, prior work (Peebles & Xie, 2023) has shown that a DiT achieves better image generation quality than a UNet with equivalent FLOPs. This prompts our curiosity to study whether a DiT can also demonstrate superiority in generalization when using the PSNR gap as a metric. Fig. 2 compares the PSNR and PSNR gap of a UNet and a DiT. Interestingly, when the number of training images is sufficient for the model size, e.g., $N=10^5$, the training and testing PSNR curves of both DiT and UNet are nearly identical, and their PSNR gaps remain small. This indicates that DiT and UNet have no substantial performance difference in distribution mapping given sufficient training data. Nevertheless, as shown in Fig. 2(b), when trained with less data, e.g., $N=10^3$ and $N=10^4$, a DiT has a remarkably smaller PSNR gap than a UNet, suggesting that a DiT has a better generalization ability than a UNet. This discrepancy of the PSNR gap motivates us to explore the underlying inductive biases that contribute to this generalization difference.

### 2.2 Eigen-Decomposition Analysis Cannot Explain DiT Generalization

Kadkhodaie et al. (2024) reveal that the generalization of a locally linear one-channel UNet is driven by the emergence of geometry-adaptive harmonic bases. These harmonic bases are obtained from the eigenvectors of a locally linear UNet's Jacobian matrix. This raises an important question: Do classic hybrid UNets and DiTs also possess harmonic bases that can account for their generalization difference? Unfortunately, due to the use of nonlinear operations such as softmax in transformer blocks and normalization layers in both convolution and transformer layers, the eigen-decomposition analysis used by Kadkhodaie et al. (2024) fails to reveal meaningful insights about the inductive

---

[2] www.github.com/openai/improved-diffusion

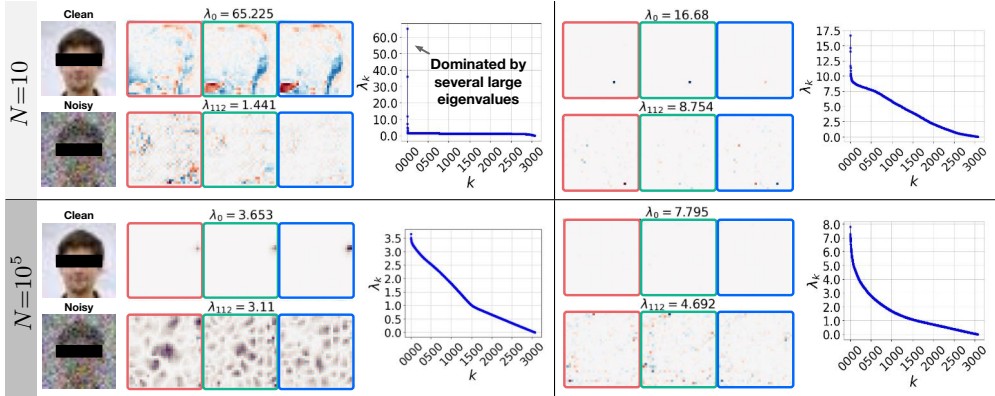

(a) UNet, FLOPs: 303.17G; Params: 109.55M   (b) DiT, FLOPs: 300.49G; Params: 14.27M

Figure 3: Jacobian eigenvector comparison between the hybrid UNet (Nichol & Dhariwal, 2021) and DiT (Peebles & Xie, 2023) with equivalent FLOPs. (a) The eigenvectors of a hybrid UNet form harmonic bases that tend to memorize the training images when $N=10$, but do not adapt well to the input geometry, differing from the behavior observed by Kadkhodaie et al. (2024). In contrast, (b) the DiT's eigenvectors do not form harmonic bases at either $N=10$ or $N=10^5$. Overall, the eigen-decomposition analysis for both the hybrid UNet and DiT fails to reveal sufficient insight into the inductive biases underlying their generalization.

biases that explain the generalization difference between a UNet and a DiT. To investigate this further, we follow Kadkhodaie et al. (2024) and perform an eigen-decomposition of the Jacobian matrices for a three-channel hybrid UNet (Nichol & Dhariwal, 2021) and a DiT. Specifically, we first feed a noisy image $x$ ($x_t$, $t$ is omitted for readability) into a DiT and a UNet and obtain their Jacobian matrices:

$$\text{Jacobian } \nabla \boldsymbol{\epsilon}_\theta = \begin{bmatrix} \frac{\partial \hat{\epsilon}_1}{\partial x_1} & \cdots & \frac{\partial \hat{\epsilon}_1}{\partial x_{HW}} \\ \vdots & \ddots & \vdots \\ \frac{\partial \hat{\epsilon}_{HW}}{\partial x_1} & \cdots & \frac{\partial \hat{\epsilon}_{HW}}{\partial x_{HW}} \end{bmatrix} \tag{5}$$

Each entry of the Jacobian represents the partial derivative of an output pixel *w.r.t.* all input pixels. Next, we perform an eigen-decomposition of the Jacobian matrix and obtain the eigenvectors.

Fig. 3 presents the eigenvalues and eigenvectors of a hybrid UNet and a DiT trained with 10 and $10^5$ images, respectively. For a UNet trained with a small dataset (*e.g.*, $N=10$), the Jacobian eigenvectors corresponding to several large eigenvalues tend to memorize the geometry of the input image. The leading eigenvalues are significantly larger than the rest, suggesting that the UNet trained with 10 images primarily memorizes training examples (Carlini et al., 2023; Somepalli et al., 2023). When the training set size increases to $N=10^5$, the UNet's eigenvectors exhibit oscillatory patterns whose frequency increases as eigenvalues $\lambda_k$ decrease. However, these harmonic bases no longer adapt well to the geometry of the input image.

In contrast, as shown in Fig. 3(b), the eigenvectors of a DiT display random, sparse patterns regardless of the training dataset size. Unlike the UNet, the eigenvalue distribution of the DiT changes little between $N=10$ and $N=10^5$, and no harmonic bases emerge. Overall, the eigen-decomposition of the Jacobian matrices for hybrid UNets and DiTs does not reveal the geometry-adaptive harmonic bases observed by Kadkhodaie et al. (2024). This does not imply that hybrid UNets and DiTs aren't capable of forming such bases. It rather indicates that the eigen-decomposition analysis is invalid for characterizing the inductive biases underlying DiT generalization. This observation motivates us to seek alternatives to investigate the inductive biases that enable the generalization of DiTs.

## 2.3 How Does a DiT Generalize?

The generalization of a DiT may originate from the self-attention (Vaswani, 2017) dynamics because of its pivotal role in a DiT. Could the attention maps of a DiT provide insights into its inductive biases? To shed light, we empirically compare the attention maps of DiTs with varying levels of

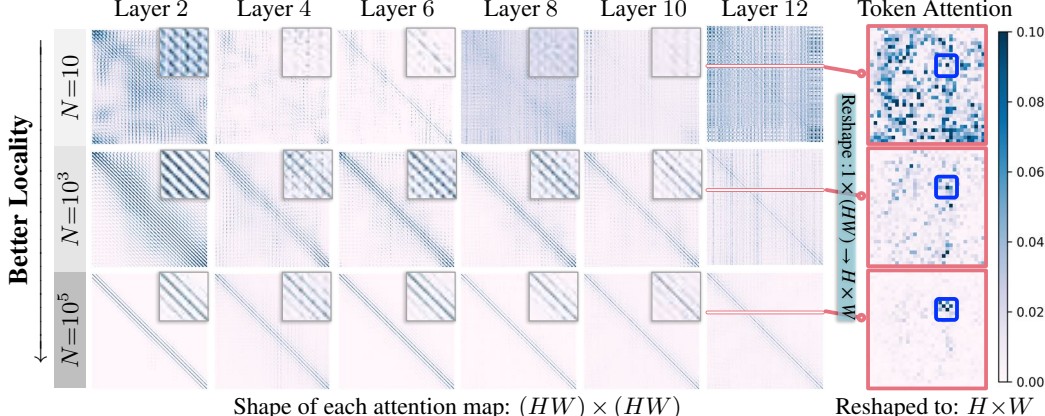

Figure 4: Attention maps of DiTs trained with $10$, $10^3$, and $10^5$ images. The top-right insets provide a zoomed-in view of the center patch of each attention map. As the number of training images increases, DiT's generalization improves, and attention maps across all layers exhibit stronger locality. The pink boxes highlight the attention corresponding to a specific output token, obtained by reshaping a single row from the layer-12 attention map (original shape: $1 \times (HW)$) into a matrix of shape $H \times W$. As $N$ increases from $10$ to $10^5$, the token attentions progressively concentrate around the region near the output token (highlighted with blue boxes).

generalization: three DiT models trained with $10$, $10^3$, and $10^5$ images, where a DiT trained with more images demonstrates stronger generalization. Specifically, we extract and visualize the attention maps from the self-attention layers of these DiT models as follows,

$$\text{Attention Map} = \text{Softmax}\left(\frac{QK^\top}{\sqrt{d}}\right), \tag{6}$$

where $\{Q, K\} \in \mathbb{R}^{(HW) \times d}$ represent the query and key matrices. $H$ and $W$ are the height and width of the input tensor, while $d$ denotes the dimension of a self-attention layer. For better readability of the attention maps, we linearly normalize each attention map to the range of $[0, 1]$ and apply a colormap to the interval $[0, 0.1]$, *i.e.*, values exceeding the upper bound are clipped at $0.1$.

Fig. 4 shows the attention maps of DiTs with varying levels of generalization on a randomly selected image. Empirically, we observe that the attention maps of a DiT's self-attention layer remain highly consistent across different images. Further details are provided in Appendix F. As the number of training images increases from $N{=}10$ to $N{=}10^5$, the attention maps of a DiT become increasingly concentrated along several diagonal lines, especially in early layers. A closer inspection of the attention values of a specific target token, *i.e.*, a row in the attention map, shows that these diagonal patterns highlight spatially close locations, indicating that the generalization ability of a DiT is linked to the locality of its attention maps.

## 3 Verifying Attention Locality as a Bias by Restricting Attention

To verify attention locality as an inductive bias, as observed in Fig. 4, we assess how much an attention map deviates from a pure identity attention. Specifically, for the attention map $\text{Attn} \in \mathbb{R}^{(M \times N)}$ corresponding to a target output token at location $(i, j)$, we compute the deviation

$$\text{Dev}(i, j) = \frac{1}{MN} \sum_{(m,n)} \left(D_{(m,n)}^{(i,j)} * \text{Attn}(m, n)\right), \text{ where } D_{(m,n)}^{(i,j)} = \sqrt{(m-i)^2 + (n-j)^2}. \tag{7}$$

Eq. (7) measures how much the attention map $\text{Attn}$ deviates from the target token at location $(i, j)$ (Wasserstein distance). We obtain the deviation for the whole attention map $\text{Attn}$ by averaging the deviation for all target tokens. In the first row of Tab. 1, we provide the deviation averaged over 300 random test images using a DiT trained with $10^3$, $10^4$, and $10^5$ images. When increasing the number of training images (*e.g.*, from $10^3$ to $10^5$), the DiT tends to generalize better, which

Table 1: Deviation↓ comparison between DiTs with and without local attention. In this setting, local attention with window sizes of $(3, 5, 7, 9, 11, 13)$ is applied to the first six layers of the DiT. $1 \times 1$ and $5 \times 5$ denote the local kernel sizes from which the attention maps deviate.

| DiT Layers | Layer 1 | | | Layer 5 | | | Layer 9 | | |
|---|---|---|---|---|---|---|---|---|---|
| **Train Set Size** | $10^3$ | $10^4$ | $10^5$ | $10^3$ | $10^4$ | $10^5$ | $10^3$ | $10^4$ | $10^5$ |
| DiT-XS/1 ($1 \times 1$) | 1.977 | 0.153 | 0.073 | 0.174 | 0.049 | 0.016 | 0.049 | 0.029 | 0.037 |
| DiT-XS/1 ($5 \times 5$) | 0.274 | 0.016 | 0.010 | 0.075 | 0.055 | 0.033 | 0.070 | 0.054 | 0.040 |
| w/ Local ($1 \times 1$) | 0.002 | 0.002 | 0.002 | 0.019 | 0.005 | 0.004 | 0.111 | 0.046 | 0.052 |

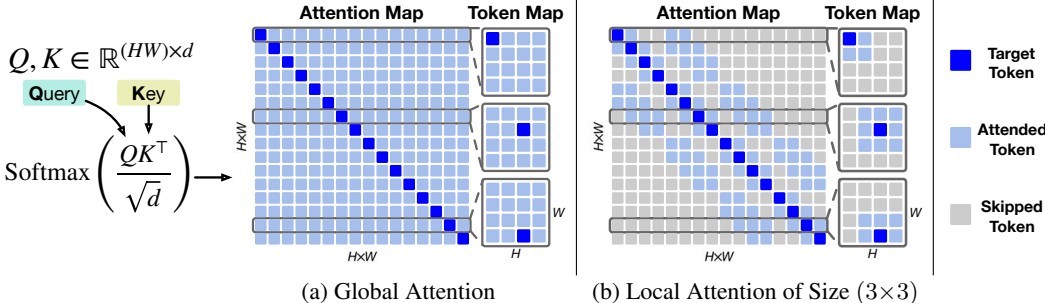

Figure 5: Global and local attention maps: (a) global attention captures the relationship between the target token and any input token, whereas (b) local attention focuses only on tokens within a nearby window around the target.

is accompanied by a reduction of the deviation, especially in early layers. A similar reduction is observed when measuring the deviation from a $5 \times 5$ local attention kernel, as shown in the second row of Tab. 1. Based on this observation, we hypothesize that it is possible to adjust the inductive bias of a DiT by restricting attention windows of early layers. To test this, we set up baselines by adopting the DiT implementations from the official repository[3] of Peebles & Xie (2023). Specifically, we remove the auto-encoder and set the patchify size to $1 \times 1$, transforming it into a pixel-space DiT. This modification rules out irrelevant components and ensures more straightforward comparisons in downstream experiments. For model training, we use images of resolution $32 \times 32$, which is equivalent in dimensionality to $512 \times 512$ for a latent-space DiT with a patchify size of $2 \times 2$.

In the remainder of this section, we show that based on the PSNR gap, injecting local attention in early layers can effectively modify a DiT generalization, often accompanied by an FID change when insufficient training data is used. Furthermore, we discover that placing the attention window restrictions at different locations in a DiT and adjusting the effective attention window sizes allows for additional control over its generalization behavior. Details *w.r.t.* experimental settings, theoretical connections to other inductive biases, more quantitative, qualitative, as well as generation results, and limitations are deferred to the Appendices.

### 3.1 Attention Window Restriction

Local attention, initially proposed to enhance computational efficiency (Liu et al., 2021; Yang et al., 2022; Hatamizadeh et al., 2023; Hassani et al., 2023), is a straightforward yet effective way to modify a DiT's generalization. Different from global attention which enables a target token to connect with all input tokens (Fig. 5(a)), local attention only permits a target token to attend within a small nearby window. The resulting attention map structure is depicted in Fig. 5(b). Notably, a local attention constrains the attention map to a sparse activation pattern only along the diagonal direction, thereby enforcing locality of the attention map. The resulting attention map patterns produced by a local attention align well with the inductive bias that a DiT exhibits when observing a strong generalization ability, as illustrated in Fig. 4 (row $N = 10^5$).

Using local attentions in early layers of a DiT can consistently improve its generalization (measured by PSNR gap) across different datasets and model sizes. Specifically, we consider a DiT model with

---
[3]https://github.com/facebookresearch/DiT

Table 2: PSNR gap↓ comparison between a DiT with and without local attention for two architectures: DiT-XS/1 and DiT-S/1. *Local* denotes applying local attention with window sizes $(3, 5, 7, 9, 11, 13)$ to the first six layers of the DiT.

| Dataset | CelebA | | ImageNet | | MSCOCO | | LSUN Church | | LSUN Bedroom | | LSUN Bridge | | LSUN Tower | |
|---|---|---|---|---|---|---|---|---|---|---|---|---|---|---|
| Train Set Size | $10^4$ | $10^5$ | $10^4$ | $10^5$ | $10^4$ | $10^5$ | $10^4$ | $10^5$ | $10^4$ | $10^5$ | $10^4$ | $10^5$ | $10^4$ | $10^5$ |
| DiT-XS/1 | 0.80 | 0.01 | 1.08 | 0.05 | 0.60 | 0.13 | 0.38 | 0.00 | 0.70 | 0.26 | 0.52 | 0.03 | 0.52 | 0.00 |
| w/ Local | 0.57 | 0.01 | 0.74 | 0.05 | 0.41 | 0.13 | 0.21 | 0.00 | 0.52 | 0.26 | 0.34 | 0.03 | 0.33 | 0.00 |
| | −0.29 | - | −0.31 | - | −0.19 | - | −0.45 | - | −0.26 | - | −0.35 | - | −0.37 | - |
| DiT-S/1 | 2.30 | 0.02 | 0.65 | 0.05 | 0.72 | 0.13 | 0.61 | 0.00 | 0.94 | 0.26 | 1.74 | 0.03 | 1.97 | 0.00 |
| w/ Local | 1.73 | 0.02 | 0.43 | 0.05 | 0.54 | 0.13 | 0.36 | 0.00 | 0.64 | 0.26 | 1.26 | 0.03 | 1.34 | 0.00 |
| | −0.25 | - | −0.34 | - | −0.18 | - | −0.41 | - | −0.32 | - | −0.28 | - | −0.32 | - |

Table 3: FID↓ comparison between a DiT with and without local attention. The best results are highlighted in **bold** font.

| Model | CelebA | | ImageNet | | MSCOCO | | LSUN Church | | LSUN Bedroom | | LSUN Bridge | | LSUN Tower | |
|---|---|---|---|---|---|---|---|---|---|---|---|---|---|---|
| Train Set Size | $10^4$ | $10^5$ | $10^4$ | $10^5$ | $10^4$ | $10^5$ | $10^4$ | $10^5$ | $10^4$ | $10^5$ | $10^4$ | $10^5$ | $10^4$ | $10^5$ |
| DiT-XS/1 | 9.69 | 2.63 | 52.57 | **17.31** | 28.35 | **12.97** | 12.88 | **4.38** | 14.84 | 5.41 | 23.18 | **8.08** | 12.55 | **4.66** |
| w/ Local | **8.46** | **2.55** | **43.87** | 18.07 | **24.43** | 13.47 | **10.48** | 4.47 | **11.96** | **5.35** | **18.15** | 8.35 | **10.56** | 4.80 |
| DiT-S/1 | 23.25 | **2.33** | 36.64 | **20.61** | 29.25 | 13.78 | 14.88 | **3.94** | 16.11 | **4.61** | 51.57 | **5.80** | 28.97 | **3.19** |
| w/ Local | **20.78** | **2.33** | **33.18** | 20.80 | **27.11** | **13.16** | **11.75** | 4.41 | **11.68** | 5.05 | **37.65** | 5.88 | **21.81** | 3.56 |

12 DiT blocks, and replace the first 6 global attention layers with local attentions, whose window sizes range from $3\times3$ to $13\times13$ with a stride of 2. We train both the vanilla DiT and a DiT equipped with local attentions with $N=10^3, 10^4$ and $10^5$ images for the same 400k training steps. Then we calculate the PSNR gap between the training and testing images for models trained with different amounts of images. In Tab. 2, we show the PSNR gap comparison between a DiT with and without local attentions on CelebA, ImageNet, MSCOCO, and LSUN (Church, Bedroom, Bridge, Tower) datasets, using baseline DiT models of two sizes (DiT-XS/1 and DiT-S/1). Notably, using local attentions reduces a DiT's PSNR gap with different amounts of training images. Importantly, the advantage of local attention is robust across different training datasets and backbone sizes.

For a discriminative model, *e.g.*, a classifier, better generalization may lead to better model performance when the training dataset is insufficient. Is this also the case for generative models like a DiT? To investigate, we compare the FID between the default DiT and a DiT using local attentions. For each dataset, we compare FID values of models trained with $10^4$ and $10^5$ images: the former represents the case of insufficient training images while the later case refers to use of sufficient training data. Tab. 3 shows the FID comparison among the same seven datasets and the two DiT backbones used when comparing PSNR gaps. Improving the generalization via local attentions can indeed improve the FID when $N=10^4$. When $N=10^5$, adding local attentions either results in comparable FID values or leads to a slight compromise because a DiT trained with sufficient data can naturally develop a local attention pattern as shown in Fig. 4. So further encouraging attention locality is expected to have limited effect. However, it offers the added benefit of reducing FLOPS with minimal performance loss. Both observations are in line with findings from discriminative models. Interestingly, we find that modifying the placement and effective attention window size permits to control a DiT's generalization and generation quality. More discussions are in Sec. 3.2 and Sec. 3.3 below. Going back to Tab. 1, the third row shows the deviation when using local attention in early layers of a DiT. As expected, using local attention reduced the deviation of early layers. Interestingly, the deviation of the remaining layers without local attention increased. This shows that other factors beyond locality of attention are at play. We leave identification and a study of those to future work.

In light of Occam's razor, reducing the model parameter count has been shown to be yet another possible strategy to inject an inductive bias. This differs from the attention window restrictions considered above, as local attentions reduce the FLOPs of a DiT without changing the model parameter count. In contrast, to inject an inductive bias by reducing the parameter count of a DiT, we explore sharing of the parameters of a DiT's attention blocks as well as modifying a DiT's attention layers to learn the coefficients of pre-computed offline PCA components. Neither of these methods showed as compelling improvements of the generalization (measured via the PSNR gap) as using local attention. We provide more details regarding the considered techniques in Appendix I.

Table 4: PSNR gap↓ and FID↓ comparison for different local attention placement patterns. The best results are highlighted in **bold** font.

| PSNR Gap | CelebA | | | ImageNet | | |
|---|---|---|---|---|---|---|
| Train Set Size | $10^3$ | $10^4$ | $10^5$ | $10^3$ | $10^4$ | $10^5$ |
| DiT-XS/1 | 7.49 | 0.80 | 0.01 | 7.77 | 1.08 | 0.05 |
| w/ Local (head) | **6.56** | **0.57** | **0.01** | **6.76** | 0.74 | 0.05 |
| w/ Local (mix) | 7.66 | 1.05 | **0.01** | 7.27 | **0.58** | 0.05 |
| w/ Local (tail) | 9.05 | 1.83 | 0.02 | 8.83 | 1.46 | 0.05 |
| w/ Local* (head) | **5.42** | **0.36** | **0.01** | **4.94** | **0.15** | 0.05 |
| w/ Local* (mix) | 6.99 | 0.86 | **0.01** | 7.12 | 0.92 | 0.05 |
| w/ Local* (tail) | 8.04 | 1.59 | 0.02 | 8.26 | 1.05 | 0.05 |

| FID | CelebA | | ImageNet | |
|---|---|---|---|---|
| Train Set Size | $10^4$ | $10^5$ | $10^4$ | $10^5$ |
| DiT-XS/1 | 9.69 | 2.63 | 52.57 | 17.31 |
| w/ Local (head) | **8.46** | 2.55 | 43.87 | 18.07 |
| w/ Local (mix) | 11.89 | 2.50 | **37.64** | 18.44 |
| w/ Local (tail) | 18.07 | **2.43** | 59.85 | **17.58** |
| w/ Local* (head) | **7.23** | 3.10 | **29.25** | 23.79 |
| w/ Local* (mix) | 10.95 | **2.71** | 51.82 | **18.80** |
| w/ Local* (tail) | 17.04 | 3.04 | 49.64 | 22.17 |

Table 5: PSNR gap↓ and FID↓ changes when the effective attention window size is kept constant, decreased, or increased. Best results are highlighted in **bold** font.

| PSNR Gap | CelebA | | | ImageNet | | |
|---|---|---|---|---|---|---|
| Train Set Size | $10^3$ | $10^4$ | $10^5$ | $10^3$ | $10^4$ | $10^5$ |
| Local Attn ($5^{*6}$) | 7.19 | 1.05 | 0.02 | **6.55** | 0.69 | 0.05 |
| ($3^{*2}, 5^{*2}, 7^{*2}$) | **7.00** | **1.01** | 0.02 | **6.55** | **0.66** | 0.05 |
| Local | 6.56 | 0.57 | 0.01 | 6.76 | 0.74 | 0.05 |
| (smaller win size) | **6.09** | **0.54** | 0.01 | **6.33** | **0.63** | 0.05 |
| Local* | **5.42** | **0.36** | 0.01 | **4.94** | **0.15** | 0.05 |
| (larger win size) | 5.92 | 0.46 | 0.01 | 6.15 | 0.56 | 0.05 |

| FID | CelebA | | ImageNet | |
|---|---|---|---|---|
| Train Set Size | $10^4$ | $10^5$ | $10^4$ | $10^5$ |
| Local Attn ($5^{*6}$) | 12.98 | **2.33** | **40.74** | 17.87 |
| ($3^{*2}, 5^{*2}, 7^{*2}$) | **12.67** | 2.35 | 40.75 | **17.75** |
| Local | 8.46 | **2.55** | 43.87 | **18.07** |
| (smaller win size) | **8.05** | 2.72 | **39.58** | 18.94 |
| Local* | **7.23** | 3.10 | **29.25** | 23.79 |
| (larger win size) | 7.88 | **2.86** | 37.87 | **19.36** |

## 3.2 Placement of Attention Window Restriction

For local attention, we study three placement schemes: 1) using local attention in early layers of a DiT, 2) interleaving local attention with global attention, and 3) placing local attention on the final layers of a DiT. In Tab. 4, we compare the PSNR gap for the three schemes on the CelebA and ImageNet data, using two distinct local attention configurations. Specifically, *Local* refers to a setting with 6 attention layers, where the window sizes vary from 3×3 to 13×13 with a stride of 2, which is consistent with the local attention configuration used in Tab. 2 and Tab. 3 above. Meanwhile, *Local** represents a different configuration consisting of 9 local attention layers, arranged as $(3^{*3}, 5^{*3}, 7^{*3})$, where $i^{*j}$ indicates repeating a local attention layer with a $(i \times i)$ window $j$ times.

The results in Tab. 4 indicate that applying local attention in the early layers of a DiT leads to a smaller PSNR gap across different training data sizes, corroborating our earlier findings. Additionally, the FID results in Tab. 4 show that the first placement scheme generally improves FID when the training data is limited ($N=10^4$). In contrast, interleaving local and global attention, or applying local attention on the final layers, enhances the model's data-fitting ability but often compromises generalization. These two placement schemes tend to improve FID when $N=10^5$ at the cost of reduced FID when $N=10^4$, further supporting the generalization results measured by the PSNR gap.

## 3.3 Effective Attention Window Size Analysis

Adjusting the effective attention window size provides an additional mechanism to control the generalization of a DiT. Specifically, our analysis reveals that smaller attention windows lead to stronger generalization, while larger windows enhance data fitting, typically at the cost of generalization. Furthermore, maintaining the total attention window size but altering the distribution across local attentions generally preserves the overall behavior of a DiT. These observations are based on an empirical study using the CelebA and ImageNet datasets, involving three paired comparisons of local attention configurations. The PSNR gap and FID results are shown in Tab. 5.

Specifically, in the first comparison, we apply two configurations of local attentions with window sizes $(5, 5, 5, 5, 5, 5)$ and $(3, 3, 5, 5, 7, 7)$ to the first six layers of a DiT. We observe that altering the attention window size distribution, while keeping the total window size fixed, has a limited impact on a DiT's generalization, as indicated by the similar PSNR gaps across $N=10^3$, $10^4$, and $10^5$. This similarity in generalization is further corroborated by their comparable FID values. In the second and third comparisons, using the DiT-XS/1 configurations with *Local* and *Local** attention settings, we

find that reducing the attention window size enhances generalization, while increasing the window size diminishes it. This is evidenced by a decrease in the PSNR gap for smaller window sizes and an increase for larger ones. Furthermore, the improved generalization is associated with better FID values under comparably insufficient training data, and vice versa.

## 4 Related Work

**Inductive Biases of Generative Models.** Current diffusion models (Sohl-Dickstein et al., 2015; Song et al., 2020; Ho et al., 2020; Kadkhodaie & Simoncelli, 2020; Nichol & Dhariwal, 2021; Song et al., 2020; An et al., 2024) exhibit strong generalization abilities (Zhang et al., 2021; Keskar et al., 2016; Griffiths et al., 2024; Wilson & Izmailov, 2020), relying on inductive biases (Mitchell, 1980; Goyal & Bengio, 2022). Prior to the emergence of diffusion models, Zhao et al. (2018) show that generative models like GANs (Goodfellow et al., 2020) and VAEs (Kingma, 2013) can generalize to novel attributes not presented in the training data. The generalization ability of generative models is often attributed to inductive biases introduced by model architecture and training (Zhang et al., 2021; Keskar et al., 2016). Kadkhodaie et al. (2024) link the generalization of diffusion models to geometry-adaptive harmonic bases (Mallat et al., 2020), but their analysis focuses on a simplified one-channel UNet. It remains unclear whether their findings extend to standard three-channel UNets (Nichol & Dhariwal, 2021) or DiTs (Peebles & Xie, 2023). This work addresses this gap: we show that UNets still exhibit harmonic bases, whereas DiTs do not. Instead, DiTs generalize through a different inductive bias – attention locality. In contrast to Zhang et al. (2024), who argue that diffusion models converge to the optimal score function largely independent of architecture, we focus on the architectural inductive biases that influence the diffusion model generalization. Recent works (Wang & Vastola, 2024; Li et al., 2024) examine the linearity of score functions but do not address architectural biases. Niedoba et al. (2024) observe that diffusion models resemble patch-based denoisers; our discovery of attention locality in DiTs offers an explanation for this behavior.

**Attention Window Restrictions.** Restricting attention windows through mechanisms such as local attention (Beltagy et al., 2020; Liu et al., 2021; Hassani et al, 2023), strided attention (Wang et al., 2021; Xia et al., 2022), and sliding attention (Pan et al., 2023), among others, can significantly improve the efficiency of attention computation (Yang et al., 2022; Hatamizadeh et al., 2023; Hassani et al., 2023; Apple, 2024). These techniques limit the attention scope, reducing computational complexity while retaining the model's ability to capture important contextual information. However, our work reveals another use for controlling the locality of attention. We show that beyond efficiency gains, local attention can be used to modulate the model's generalization by enforcing the inductive bias of locality within attention maps. We think this is particularly important for science domains where data for training generative models is less abundant.

## 5 Conclusion

This paper investigates the inductive biases that facilitate the generalization ability of DiTs. For insufficient training data, we observe that DiTs achieve superior generalization, as measured by the PSNR gap, compared to UNets with equivalent FLOPs. However, the eigen-decomposition analysis that reveals geometry-adaptive harmonic bases as the key inductive bias of diffusion models based on locally linear UNets becomes invalid for classic hybrid UNets and DiTs due to the presence of nonlinear operations. Therefore, we take an alternative approach to explore alternative inductive biases and identify that a DiT's generalization is instead influenced by the locality of its attention maps. Consequently, we effectively modulate the generalization behavior of DiTs by incorporating local attention layers. Specifically, we demonstrate that varying the placement of local attention layers and adjusting the effective attention window size enables fine-grained control of a DiT's generalization and data-fitting capabilities. Enhancing a DiT's generalization often leads to improved FID scores when trained with insufficient data. One limitation of this work is that our analysis focuses exclusively on DiTs. For future work, we consider it important and interesting to study the generalization behavior of hybrid models and conditional transformers (*e.g.*, MMDiT modules), given their growing popularity in recent generative architectures.

## Acknowledgement

We sincerely thank Zhongzheng Ren, Chen Chen, Byeongjoo Ahn, Saeed Khorram, and Aditya Sankar for their valuable discussions and insightful feedback. We are also deeply grateful to Alex Colburn, Qi Shan, and the Video Computer Vision team at Apple for their generous support in providing the infrastructure and computational resources that made our experiments possible.

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
