# OpenReview forum: "On Inductive Biases That Enable Generalization in Diffusion Transformers"
_NeurIPS.cc/2025/Conference — NeurIPS 2025 poster_

### Official Review · Reviewer_dzs6 · 2025-06-04

**Clarity:** 3
**Significance:** 3
**Originality:** 3
**Rating:** 5
**Confidence:** 3

**Summary:**

This paper studies the inductive bias of DiT models. It is found that the inductive bias of DiT models does not manifest as geometry-adaptive harmonic bases as UNet models do. Instead, the authors discover that it is related to the locality of early self-attention layers. Furthermore, abundant experiments validate this finding.

**Questions:**

What could happen if the attention window is somehow enlarged in DiTs trained with sufficient data? Could this modification make them generalize worse?

**Ethical Concerns:**

["NO or VERY MINOR ethics concerns only"]

**Final Justification:**

I have read the response and the discussion between the authors and other reviewers. It seems there is very mixed opinion. Nevertheless, I insist that this paper as an exploratory study can reveal some interesting properties of DiT models, even though it as a practical method for improving DiT generation might not be that well applicable. I will keep my rating not changed as it is already high enough.

**Limitations:**

I recommend considering the special ViT-UNet-mixed architecture of Stable Diffusion XL as a possible future topic.

**Paper Formatting Concerns:**

No.

**Quality:**

3

**Strengths And Weaknesses:**

Pros:
1. The authors are focusing on the newest trend of diffusion model advancement, studying the property of DiT models.
2. This study has a solid methodology, along with abundant experimental evidence and an interesting finding.
3. For certain scenarios where the training data is scarce, the finding might have some practical impact.

Minor Cons:
1. The architecture of actual commercial diffusion models is more complex than the simplified versions in this study. For example, Stable Diffusion XL model, as a UNet model, has many 10-layer ViTs integrated in the UNet, which may make it behave specially. In Stable Diffusion 3 and Flux, the attention blocks are upgraded into MMDiT fusion attention, which may also introduce new properties. I would recommend some future work on commercial diffusion models to extend the study to more complex architectures.

---

> ### Author Rebuttal · Authors · 2025-07-31
>
> Thank you very much for your time, thoughtful feedback, and for highlighting the strengths of our work, specifically our focus on timely advances in diffusion models, the solid methodology, the well-supported experimental results, and the interesting findings. We also appreciate your recognition of the potential practical value of our method in low-data regimes. We address the remaining questions below.
>
> ---
> * **Q1: Suggestion to extend the analysis to more complex commercial diffusion architectures (e.g., SDXL, SD3, Flux) which include hybrid UNet and ViTs and conditional attention blocks.**
>
>   Thanks for the valuable suggestion. As stated in line 40-42 of the appendix, the UNet architecture used in our experiments contains one attention layer at the last encoder stage and one attention layer at the middle block, which is a hybrid model combining convolution and two attention layers. In general, we agree that studying the generalization behaviour of hybrid models and conditional transformers (*e.g.*, MMDiT modules) are important and interesting due to the increasing popularity of these architectures.
> ---
> * **Q2: What could happen if the attention window is somehow enlarged in DiTs trained with sufficient data? Could this modification make them generalize worse?**
>
>   For attention layers of a normal DiT, each token has full access to all other tokens, meaning that the attention window size is already at the maximum and cannot be enlarged. On the other hand, a DiT trained with sufficient data already forms a locality pattern in its attention layers as shown in Fig. 4 ($N=10^5$ row). Therefore, we think further enlarging the attention window would have very limited impact on a DiT’s generalization and performance since newly added non-local tokens in attention don’t contribute much.
> ---
> * **Q3: I recommend considering the special ViT-UNet-mixed architecture of Stable Diffusion XL as a possible future topic.**
>
>   Thank you for a valuable suggestion. We agree and will note these future directions in the revised paper.

---

> > ### Comment · Reviewer_dzs6 · 2025-08-01
> >
> > I have read the response and the discussion between the authors and other reviewers. It seems there is very mixed opinion. Nevertheless, I insist that this paper as an exploratory study can reveal some interesting properties of DiT models, even though it as a practical method for improving DiT generation might not be that well applicable. For now, I will keep my rating not changed.

---

### Official Review · Reviewer_jHWX · 2025-06-06

**Clarity:** 2
**Significance:** 2
**Originality:** 3
**Rating:** 3
**Confidence:** 3

**Summary:**

This paper studies the inductive biases of DiT for generalization. The research question is: do transformer-based denoising networks have inductive biases that can be expressed via geometry-adaptive harmonic bases like in UNet-based denoisers? The paper then shows that this is not the case, and proposes to restrict the attention windows in early layers, which can improve the generalization ability. The paper conducts experiments on CelebA. ImageNet, MSCOCO, and LSUN datasets, using DiT.

**Questions:**

it might be a bit narrow if the paper only focuses on geometry-adaptive harmonic bases as in UNet-based denoisers. Is it possible to study the generalization of DiT through a more general exploration?

**Ethical Concerns:**

["NO or VERY MINOR ethics concerns only"]

**Final Justification:**

Thanks the authors for rebuttal. My concerns on performance have been addressed. However, in my opinion the scope and significance of this work is still a bit limited and specific to certain DiT models. Thus I keep my score unchanged.

**Limitations:**

yes

**Quality:**

2

**Strengths And Weaknesses:**

strength:
1. the research question is specific and clearly stated.

2. the analysis in section 2.3 and the verification in section 3 is reasonable.

weakness:
1. it seems that the result is not very promising. As shown in Table 3, using the proposed local attention can be harmful to the performance in the 10^5 setting. Although it can improve the performance in the 10^4 setting, the FID is still high. This makes the applicability of this work a bit questionable.

2. The proposed method is not very easy to be adapted to many other Transformer-based Diffusion models. The effectiveness of the method is affected by window sizes and which layers being modified. If we target on other Transformer-based models, does it mean that we should do these analysis again and try to choose appropriate parameters?

---

> ### Author Rebuttal · Authors · 2025-07-31
>
> Thanks a lot for your time, feedback, and highlighting that our research question is specific and clearly stated while our analysis in Sec. 2.3 and verification in Sec. 3 are reasonable. We answer the remaining questions below.
>
> ---
> * **Q1: It seems that the result is not very promising. As shown in Table 3, using the proposed local attention can be harmful to the performance in the $10^5$ setting. Although it can improve the performance in the $10^4$ setting, the FID is still high. This makes the applicability of this work a bit questionable.**
>
>   This paper aims to analyze the inductive bias of a DiT that contributes to its generalization, as stated in line 81-86. Based on experiments reported in Sec. 2, we hypothesize that attention locality is an important indicator of generalization. Our experiments hence modify attention patterns by enforcing attention locality while assessing the resulting impact on the generalization of a DiT (FID). This should not be interpreted as a new method, but rather as studies to verify that the hypothesized attention locality bias is indeed at play. Regarding the FID results shown in Tab. 3, when trained with $10^5$ images, as shown in Fig. 4 (N=$10^5$ row), attention locality patterns emerge naturally when training. Hence, it is expected that additionally encouraging attention locality doesn’t lead to any changes in a DiT’s generalization as validated via the PSNR gap in Tab. 2. For FID, we observe a minimal $4.33$% performance fluctuation averaged over seven datasets. Considering that using local attention offers the added benefit of reducing FLOPs, we think it is inaccurate to say local attention is harmful when training a DiT with $10^5$ samples. When using $10^4$ training samples, our experiments consistently show that introducing local attention in early DiT layers improves generalization (PSNR gap in Tab. 2 and Cosine Similarity in Tab. 3 of appendix) and sample quality (FID in Tab. 3, in appendix: IS in Tab. 4, FD-Dinov2 in Tab. 6). In Tab. 3, we observe a consistent $17.59$% FID improvement averaged over seven datasets, which we think is significant compared with the $4.33$% fluctuation when $N=10^5$.
> ---
> * **Q2: The proposed method may not be easy to be adapted to other Transformer-based diffusion models, given sensitivity to window size and layer selection.**
>
>   Our experimental results in Tab. 2 and 3 consider two DiTs of different sizes and seven datasets. Using local attention at beginning layers of a DiT with gradually increasing attention window size consistently improves the model generalization. In Tab. 8 of the appendix, we apply the same local attention setting for a latent-space DiT, which also demonstrates reasonable generalization and performance improvement when trained with $N=10^4$ samples, as measured by PSNR gap and FID, respectively. Given these consistent improvements, we think our default setting would work reasonably well for most DiTs. Despite this, a careful analysis of the generalization behaviour is still beneficial for a specific DiT architecture, which we consider one of the key takeaways of this paper.
> ---
> * **Q3: It might be a bit narrow if the paper only focuses on geometry-adaptive harmonic bases as in UNet-based denoisers. Is it possible to study the generalization of DiT through a more general exploration?**
>
>   This paper studies the generalization of DiTs from different perspectives. In Sec 2.2, we use a Jacobian analysis to study the representation behaviour of DiTs. Beyond that, in Sec. 2.3, we demonstrate the attention locality bias that is associated with a DiT’s generalization, which is verified by an improved PSNR gap when $N=10^4$ after injecting local attention into a DiT (Sec. 3). Further, in Appendix B, we provide a theoretical analysis connecting the discovered attention locality bias with the simplicity bias (Appendix B.1), and subsequently, low sensitivity bias (Appendix B.2), which indicates good generalization of a Transformer (Appendix B.3). Despite these, we recognize a more general exploration of a DiT’s generalization, an important and interesting future work.

---

> > ### Comment · Reviewer_jHWX · 2025-08-05
> > **Thanks for the rebuttal**
> >
> > Thanks the authors for rebuttal. My concerns on performance have been addressed. However, in my opinion the scope and significance of this work is still a bit limited and specific to certain DiT models. Thus I keep my score unchanged.

---

> > > ### Author Response · Authors · 2025-08-06
> > >
> > > Thank you for participating in the author-reviewer discussion and for confirming that we have addressed your question on experimental results when training a DiT with $10^5$ samples. We address the remaining question below.
> > >
> > > ---
> > >
> > >   We respectfully disagree with the assessment that our work has limited scope or significance. While it is commonly believed that DiTs outperform UNets due to their global attention mechanisms, our study reveals a surprising counterpoint: DiTs that generalize well tend to exhibit locality patterns in their attention maps, resembling adaptive convolutions, even though they have access to global tokens. To the best of our knowledge, this is the first work to systematically study the generalization behavior of DiTs and uncover this locality bias, thereby challenging prevailing assumptions about how DiTs function and why they perform well. This contribution has been explicitly acknowledged by Reviewers `XEv5` and `dzs6`.
> > >
> > >   Regarding scope, prior work by Kadkhodaie *et al.* [1] on UNet generalization focuses on a shallow, locally linear, three-layer UNet evaluated primarily on the CelebA dataset. In contrast, our study investigates two DiT architectures (DiT-S/1 and DiT-XS/1), spans seven datasets, covers both pixel-space and latent-space DiTs, and includes comparisons against two UNets and fourteen DiTs (see Appendix Table 1). We believe this reflects a substantially broad scope of our paper relative to existing studies.
> > >
> > >   While our study focuses on a standard DiT architecture, we think that such a targeted analysis is a necessary first step. Since it remains unclear how even a standard DiT generalizes, it is premature to draw conclusions about broader variants. We think a clear understanding of the baseline architecture is essential before extending the analysis to different kinds of DiTs.
> > >
> > >   We also noticed that the reviewer rated the paper with the lowest score on clarity (1), while other reviewers found the clarity to be good or excellent. We would be happy to clarify any parts that may have been unclear and ensure our main claims and contributions are communicated more effectively.
> > >
> > >
> > >   [1] Generalization in diffusion models arises from geometry-adaptive harmonic representation. In ICLR, 2024.

---

### Official Review · Reviewer_VB1R · 2025-06-28

**Clarity:** 3
**Significance:** 3
**Originality:** 2
**Rating:** 3
**Confidence:** 4

**Summary:**

This paper investigates the inductive biases that facilitate the generalization ability of DiT. The authors propose that locality of attention maps in a DiT’s early layers are closely associated with generalization. They use local attention windows in early layers of a DiT and observe an improvement in generalization. Experimental results on the CelebA, ImageNet, MSCOCO, and LSUN data show that strengthening the inductive bias can improve both generalization and generation quality when less training data is available.

**Questions:**

1. The experiments are not sufficient to support the proposed viewpoints.
2. Are there some generated images to demonstrate?
3. Are there some comparison results with SOTAs?

**Ethical Concerns:**

["NO or VERY MINOR ethics concerns only"]

**Final Justification:**

I appreciate the authors’ detailed response, which has clarified part of my concerns. I remain some cautious about the novelty and effectiveness. I increase my score to Borderline reject.

**Limitations:**

yes

**Quality:**

2

**Strengths And Weaknesses:**

Strengths：
1. The paper proposes the locality of attention maps as a key inductive bias contributing to the generalization of a DiT.
2. The paper demonstrates how to control this inductive bias by incorporating local attention windows into early layers of a DiT.

Weaknesses:
The experimental part has limitations. The proposed approach lacks comprehensive comparisons with state-of-the-art methods and in-depth ablation studies to fully validate both generalization and generation quality. For DiT, there is not any generated image as experiment result to demonstrate and compare. That is not enough only using PSNR gap and FID to verify the effectiveness of the viewpoints.

---

> ### Author Rebuttal · Authors · 2025-07-31
>
> Thanks a lot for your time, feedback, and highlighting that this paper reveals the locality of attention maps as an inductive bias of a DiT, which is verified by our experiments incorporating local attention windows into early layers of a DiT. We answer the remaining questions below.
>
> ---
> * **Q1: More comprehensive comparisons with state-of-the-art methods and in-depth ablation studies to fully validate both generalization and generation quality.**
>
>   This paper aims to analyze the inductive bias of a DiT that contributes to its generalization, as stated in line 81-86. Based on experiments reported in Sec. 2, we hypothesize that attention locality is an important indicator of generalization. Our experiments hence modify attention patterns by enforcing attention locality while assessing the resulting impact on the generalization of a DiT. This should not be interpreted as a new method, but rather as studies to verify that the hypothesized attention locality bias is indeed at play. To our best knowledge, there are no state-of-the-art methods to compare to which aim to control the generalization of a DiT. Nonetheless, in Appendix I, we compare the effect of encouraging attention locality with a known way of improving generalization – reducing trainable parameters. As shown in Tab. 11 of Appendix I, two parameter reduction approaches, i.e., using parameter sharing and composing attention maps with pre-defined PCAs, are not as effective in explaining a DiT’s generalization as the local attention study. Further, in Tab. 4, we show that encouraging attention locality at the first few layers of a DiT can improve its generalization. Encouraging attention locality at the last few layers or interleaving global and local attention do not. In Tab. 5, we show that use of a smaller attention window size can lead to a larger generalization improvement while a larger window size leads to a smaller improvement. All these ablation studies verify that encouraging attention locality impacts a DiT’s generalization, corroborating our hypothesis.
> ---
> * **Q2: For DiT, there is no generated image to demonstrate and compare.**
>
>   We present qualitative results in Fig. 2 of Appendix D. We find that DiTs trained with $10^4$ and $10^5$ images while using local attention produce images that are more like each other than DiTs trained without using local attention. Further, in Tab. 9 of Appendix D, we quantitatively verify that the average pixel intensity between generated images of models trained with $10^4$ and $10^5$ images is smaller if local attention is encouraged. This phenomenon shows that applying local attention to a DiT trained with $10^4$ images can make it generate images closer to a well-generalized model trained with $10^5$ images, indicating generalization improvement.
> ---
> * **Q3: Only using PSNR gap and FID to verify the effectiveness of the viewpoints is not enough.**
>
>   We consider five evaluation metrics: PSNR gap (Tab. 2), FID (Tab. 3), IS (Tab. 4 in Appendix C.3), FD-Dinov2 (Tab. 5 in Appendix C.4), and Cosine Similarity to the nearest training image (Tab. 2 in Appendix C.2). Prior works studying diffusion model generalization [1,2] only use PSNR gap and Cosine Similarity as evaluation metrics. We hence believe the metrics used in our evaluation extend prior work and comprehensively support our claim.
> ---
>   [1] Generalization in diffusion models arises from geometry-adaptive harmonic representation. In ICLR, 2024.
>
>   [2] The emergence of reproducibility and consistency in diffusion models. In ICML, 2024.

---

> > ### Comment · Reviewer_VB1R · 2025-08-07
> >
> > Could you theoretically explain why strengthening the attention locality in the early layers of DiT can improve its generalization ability?

---

> > > ### Author Response · Authors · 2025-08-08
> > >
> > > Thank you for participating in the author-reviewer discussion. We address the new question regarding a theoretical justification of our findings below.
> > >
> > > ---
> > >
> > > We provide a theoretical justification in Appendix B, connecting our findings with theoretical work [1,2,3], showing that local attention reduces the degree of monomials in the network representation, which lowers sensitivity to data perturbations – an effect linked to better generalization. More specifically, our theoretical explanation is based on prior work about the simplicity bias of transformers [1]. Preliminaries are introduced in Appendix B.1. Subsequently, we connect prior work [1] to our results in Appendix B.2 and show that local attention can encourage the simplicity bias, resulting in low sensitivity *w.r.t.* data perturbation of a transformer. Finally, in Appendix B.3, we demonstrate that low sensitivity of a transformer is connected with the existence of flat minima, which is a widely accepted indicator of good generalization of a model.
> > >
> > > ---
> > >
> > > [1] Bhavya Vasudeva, Deqing Fu, Tianyi Zhou, Elliott Kau, Youqi Huang, and Vatsal Sharan. Simplicity bias of transformers to learn low sensitivity functions. arXiv preprint arXiv:2403.06925, 2024.
> > >
> > > [2] Ronald De Wolf. A brief introduction to fourier analysis on the boolean cube. Theory of Computing, 2008.
> > >
> > > [3] Greg Yang and Hadi Salman. A fine-grained spectral perspective on neural networks. arXiv preprint arXiv:1907.10599, 2019.

---

### Official Review · Reviewer_XEv5 · 2025-07-01

**Clarity:** 4
**Significance:** 4
**Originality:** 3
**Rating:** 4
**Confidence:** 4

**Summary:**

This paper aims to understand and explain the inductive biases responsible for generalization in transformer-based diffusion models. The authors show that the underlying mapping is not harmonic, unlike UNet-based diffusion models. They propose that the locality of attention maps lead to generalization. The claim is supported with extensive experiments.

**Questions:**

- Do you observe the strong generalization (sampling the same image by two networks trained on disjoint dataset) in DiT? I understand this needs lots of compute, so I'm not asking to do the experiment during the rebuttal, but curious if you have done the experiments already.

**Ethical Concerns:**

["NO or VERY MINOR ethics concerns only"]

**Final Justification:**

I think this paper is making a significant and valuable contribution regarding generalization in DiT models. However there are some analysis regarding using a first order approximation to analyze the network that seems incorrect to me, due to the inherent local nonlinearity of the model. Although the authors agreed the analysis was incorrect, and potentially misleading, they didn't make it clear that the paper would be revised to remove or replace such analysis from the paper. So I lowering my rating from 5 to 4 because I remain unsure if the incorrect conclusions will be included in the final revision or not. That said, I still regard the paper as an important contribution to the study of locality in transformers, which justifies the relatively high score.

**Limitations:**

- No negative societal impacts.
- Yes limitations have been addressed

**Quality:**

3

**Strengths And Weaknesses:**

**Main strength:**
- I found the results in figure 4 amazing! This figure shows more than mere locality of the attention maps in the generalization regime. The plots show adaptive convolutional layer maps! It seems that with enough training data, the attention block in the transformer converges to convolutional layer. And the effective kernel size increases as the layer number increases. This is consistent with results presented in Fig 2 which shows that denoising performance is quite comparable between convolutional model and transformer based model. I find this result quite fascinating since it challenges the commonplace belief that DiT is superior to conv nets due to its access to global dependencies. The authors show the opposite! That in the generalization regime, the local information is what's needed and used even when access to global dependencies are provided.

Consider relating your results to this paper that is related but for classification: Maithra Raghu et al, Do Vision Transformers See Like Convolutional Neural Networks?, 2021

**Main weakness:**
- The Jacobian analysis is not valid as a tool for understanding the mapping in the DiT, making Fig 1 and Figure 3 difficult to interpret. The Jacobian is a first-order Taylor approximation to the denoising mapping, so it can fully explain the mapping only when the network is *locally linear*. In Kadkhodaie et al 2024, the UNet is modified *to become  locally linear*, as a result the Jacobian times the input is exactly equal to the end to end network operation (important pieces are: using ReLU, removing the bias terms of the convolutions and also using batch-norm without mean). Previous work has shown these changes do not compromise the performance but makes the network locally linear, which lends it to eigen analysis of the Jacobian. The standard DiT architecture is far from locally linear, due to the use of non-linearities other than ReLu, like softmax, and LayerNorm or GroupNorm layers. So it is not clear what a first order approximation captures about the mapping! One would need the higher orders and the affine bias to get the full picture. So  DiT may or may not be arriving at geometry adaptive harmonic bases, but the presented experiments do not answer this question. However, given the results in Fig 4, my guess is that DiT is probably arriving at GAHBs since the attention layers seems to be converging to convolutional patterns in generalization regime.

*************************
**Other points:**
- The eigen vectors associated with Nichole & Dhariwal 2021 shown in Fig 1 do not really show GAHBs despite the caption. If the original architecture is used, it contains group norm, nonlinearity other than Relu (I think Gelu) that makes the mapping locally non-linear, and I believe that's the reason.
- More importantly, the architecture used in Nichole & Dhariwal 2021 contains **attention blocks** (to my knowledge), yet the eigen vectors are very different from panel (c) in Fig1. This is another evidence that the Jacobian analysis for locally non-linear mappings is not very informative. Just from the fig 1 results, I infer that DiT is more locally non-linear than Nichole & Dhariwal 2021, maybe due to softmax and other pieces I am not aware of. But the general point is that the fig is not reflecting the true bases the networks are arriving at.
- Minor point: Labeling the UNet in Kadkhodaie et al 2024 as simplified might be misleading. The change from grayscale (one-channel input) to color (three-channel input) is not significant and doesn't impact the end to end mapping, due to the high correlation between RGB channels. In other words, the shape information is way more important and difficult to capture than color information. On the other hand, the network used in Nichol & Dhariwal 2021 could be labeled as augmented UNet since it has attention block t-conditioning, etc hence more significantly different from original UNet.
- To compute the average PSNR in eq 4, the sum should be outside of log (the two operations do not commute). i.e compute individual PSNR for each image and then take the average.
- Suggestion: replacing the deviation from identity eq 6 with deviation from convolutional kernel with a certain kernel size.
*************************
I think overall the paper is quite significant in shedding light on the operation of attention blocks in generalization regime. So despite the erroneous Jacobian analysis, I think this paper can have an impact. I would increase my score if the comments and questions about the weakness (regarding Jacobian analysis) are addressed.

---

> ### Author Rebuttal · Authors · 2025-07-31
>
> Thanks a lot for your time and feedback. We are encouraged that our findings about DiT’s locality bias shown in Fig. 4 are recognized: A DiT can form a locality pattern similar to adaptive convolution when it achieves a good generalization, despite that access to global tokens is provided. We also appreciate the suggested related work by Raghu *et al.*, which echoes our finding but in ViTs. We will discuss this new related work in the revised paper.  We answer the remaining questions below.
>
> ---
> * **Q1: DiT lacks local linearity due to the use of GeLU, Softmax, and group norm layers. Hence, the Jacobian analysis in Figs. 1 and 3 is not a valid interpretation tool for DiT’s mappings.**
>
>   We agree, a DiT is not locally linear. Note, we use the Jacobian to analyze the model’s sensitivity and representation behavior instead of linearly decomposing the full mapping. Therefore, we do not draw strong conclusions from Fig. 1 and Fig. 3 alone. Rather, we view them as suggestive empirical evidence that first-order Jacobian analysis may not suffice to study the inductive bias of a DiT. This motivates direct visualization of the attention maps of a DiT in Fig. 4, where we observe that attention patterns in DiT converge to localized structures. We’ll clarify in a revised version by adding this discussion and noting that this doesn’t rule out GAHBs.
> ---
> * **Q2: The Jacobian analysis in Fig. 1 is not as informative due to the non-linearity in both DiT and UNet used by Nichol & Dhariwal 2021.**
>
>   Thanks a lot for a great argument. To confirm, we use one attention layer at the last encoder stage and in the middle block in the UNet by Nichol & Dhariwal 2021, which is stated in line 40-42 of the appendix. We also use group norm and SiLU layers in the UNet. We think the amount of non-locality introduced by network operators is a reasonable and interesting explanation for the Jacobian eigenvector differences between the UNets and DiT in Fig. 1. We will add this discussion to Sec. 2.2 of the revised paper to provide more insight. Furthermore, for a DiT with the same FLOPs as a UNet, whether the amount of non-linearity is a deciding factor for their generalization difference is another interesting future direction.
> ---
> * **Q3: Minor: Labeling the UNet in Kadkhodaie et al. (2024) with “simplified” is misleading. The UNet used by Nichol & Dhariwal 2021 should be labeled as augmented UNet.**
>
>   Thanks for the valuable suggestion. Initially, we considered the UNet by Nichol & Dhariwal 2021 as the "standard" UNet as it has been widely used. For clarity, in the revised paper, we will rename the UNet used by Kadkhodaie et al. (2024) as the locally-linear UNet and the UNet by Nichol & Dhariwal 2021 as a hybrid UNet, recognizing that it contains attention layers.
> ---
> * **Q4: To compute the average PSNR in Eq 4, the sum should be outside of log (the two operations do not commute). i.e compute individual PSNR for each image and then take the average.**
>
>   Thank you for pointing this out. Upon reviewing our code, we confirmed that the reported PSNR values are computed by first calculating the PSNR for each image individually and then averaging the results. We will revise Eq. (4) to reflect this accordingly.
> ---
> * **Q5: Suggestion: replacing the deviation from identity in eq 6 with deviation from convolutional kernel with a certain kernel size.**
>
>   Thanks for a valuable suggestion. Below we present the deviation score *w.r.t.* a convolution kernel of $5\times5$ for a DiT-XS/1 trained with CelebA data using $10^3$, $10^4$, and $10^5$ images, respectively.
>
>   | Train Data Num | $10^3$ | $10^4$ | $10^5$ |
>   | --- | --- | --- | ---- |
>   | Layer 1| 0.274 | 0.016 | 0.010 |
>   | Layer 5| 0.075 | 0.055 | 0.033 |
>   | Layer 9| 0.070 | 0.054 | 0.040 |
>
>   The results above also show that the deviation score *w.r.t.* a convolution kernel decreases when a DiT has better generalization, *i.e.*, when it is trained with more images. We will add the new deviation score results into the revised appendix.
> ---
> * **Q6: Do you observe strong generalization (sampling the same image by two networks trained on disjoint datasets) in DiT?**
>
>   Yes, we did. We will add images generated by two DiTs trained with disjoint data into the revised appendix.

---

> > ### Comment · Reviewer_XEv5 · 2025-08-06
> >
> > Thank you for your detailed responses to my questions. I found the answers to Questions 3 through 6 clarifying, and I believe the suggested modifications will improve the paper.
> >
> > However, I remain unsatisfied with the responses to Questions 1 and 2, which I see as addressing the **main weakness** of the current version. I’d like to clarify my concern further.
> >
> > A central part of the paper is based on Jacobian analysis of the DiT model. The claim is that DiT—due to its attention blocks—does not give rise to adaptive harmonic bases, unlike fully convolutional models. This framing appears prominently: it's mentioned in the abstract and visually emphasized in Figure 1, which appears before the abstract and serves as a major conceptual anchor for the paper.
> >
> > As I stated in my review, this claim is **not** supported by the analysis presented. Due to the local nonlinearity of DiT, no conclusions can be drawn from the Jacobian about the presence or absence of adaptive harmonic structure. In your rebuttal, you write that "we view them as suggestive empirical evidence that first-order Jacobian analysis may not suffice to study the inductive bias of a DiT." But this interpretation does not match the framing in the paper. In fact, it seems to contradict it. I find this inconsistency confusing and concerning.
> >
> > My suggestion—then and now—is that the Jacobian analysis should be removed from the paper. It does not support the claim being made, and its current placement risks misleading readers. By putting an unsupported analysis and a corresponding figure front and center, the paper presents a message that is not grounded in the actual evidence.
> >
> > That said, I find the second part of the paper to be valuable, important, and novel. If the revision focuses on the locality results, I believe the paper would be significantly strengthened. You can motivate this section by noting, honestly, that Jacobian-based analysis is not feasible here due to the nonlinearities inherent in DiT. Provided the revision is made as suggested—removing the unsupported Jacobian analysis and refocusing the paper around the locality results—I stand by my recommendation for acceptance.

---

> > > ### Author Response · Authors · 2025-08-06
> > >
> > > Thank you for participating in the author-reviewer discussion and for confirming that we have addressed Questions 3 to 6. Regarding Questions 1 and 2, we sincerely appreciate your suggestions and will revise the paper accordingly.

---

### Note · Authors · 2025-08-15

Thanks for the valuable comments. We thank reviewers `XEv5` and `dzs6` for recognizing our discovered locality bias as an interesting finding challenging the common belief about how DiT works. We next restate our contribution and summarize responses to questions without further feedback to assist the AC and reviewers in their evaluation.

Contribution: This paper identifies attention locality as the inductive bias enabling diffusion transformers (DiTs) to generalize, shows that enhancing locality affects generalization, and theoretically links this bias to the low-sensitivity bias described in prior work.

* Reviewer `VB1R`
  * **Q1: Comparisons/ablations**: We clarified that our work tests the hypothesis that attention locality drives DiT generalization rather than proposing a new method. Since no prior work explicitly controls DiT generalization, we compared with parameter-reduction methods and found them less effective. Ablations show early-layer local attention with smaller windows improves generalization, whereas late-layer or interleaved local attention does not.
  * **Q2: Qualitative results**: We highlighted Appendix D: local attention produces images similar to those from well-generalized models.
  * **Q3: Evaluation metrics**: We used five metrics: PSNR gap, FID, IS, FD-Dinov2, and Cosine Similarity, covering generalization and sample quality. This extends beyond the two metrics of prior work.
  * **Q4: Theoretical explanation**: We pointed to Appendix B connecting our findings to the simplicity bias of transformers.
* Reviewer `jHWX`
  * **Q2: Adaptability to other DiTs**: Local attention works consistently across two DiT sizes, seven datasets, and a latent-space DiT.
  * **Q3: Broader exploration**: We hypothesize and examine the attention locality bias and its theoretical links to simplicity and low-sensitivity biases.
  * **Q4: Scope and significance**: We disagree with the claim that our work is narrow: first systematic study of DiT generalization while challenging the assumption that global attention is key. We consider multiple UNets, both pixel-space and latent-space DiTs, seven datasets, and extensive comparisons with five metrics, broader than prior UNet-focused studies. We also explain: focusing on a standard DiT is a necessary first step.

In summary, we believe we have comprehensively addressed all questions. We think our work offers valuable insights for generative modeling and contributes to a deeper understanding of DiT behavior.

---

### Decision · Program_Chairs · 2025-09-17

**Decision:**

Accept (poster)

**Comment:**

This paper studies the inductive biases that are responsible for generalization in diffusion models. In particular, their main finding is that DiT's trained on larger datasets exhibit local attention patterns (i.e. they attend to a local neighbourhood), while those trained on smaller datasets do not have the locality bias and hence exhibit worse generalization performance. They demonstrate this by showing that enforcing locality in the smaller datasets leads to improved performance. There was large variance in the review scores, but I agree with Reviewer XEv5 and Reviewer dzs6 that the extensive experimentation & findings are interesting and worthy of acceptance. The authors should take on Reviewer XEv5's suggestions regarding softening the claims around the Jacobian analysis (as discussed in the rebuttal period).